# NLRP6 Induces Lung Injury and Inflammation Early in Brucella and Influenza Coinfection

**DOI:** 10.3390/jpm12122063

**Published:** 2022-12-14

**Authors:** Bochang Shi, Hui Han, Huabin Li, Lingyun Tan, Xinyu Li, Keyu Wang, Bo Li, Wei He, Chongyu Tian, Fang Yan, Yanchun Shi, Yuanqiang Zheng, Zhongpeng Zhao

**Affiliations:** 1Inner Mongolia Key Laboratory of Molecular Biology, Inner Mongolia Medical University, Hohhot 010000, China; 2Qingdao Binhai University, Qingdao 266000, China; 3College of Veterinary Medicine, Shanxi Agricultural University, Jinzhong 030600, China; 4School of Basic Medicine Sciences, Anhui Medical University, Hefei 230000, China; 5Department of Clinical Laboratory, The Second Medical Center of Chinese PLA General Hospital, National Clinical Research Center for Geriatric Diseases, Beijing 100000, China; 6The Fifth Medical Center of PLA General Hospital, Beijing 100000, China; 7Beijing University of Chinese Medicine, Beijing 100000, China

**Keywords:** coinfection, Brucella, Influenza A virus, NLRP6, IL-18

## Abstract

(1) Background: With the resurgence of brucellosis epidemics in China in recent years, the chances of a brucella coinfection with other common respiratory pathogens, such as the influenza virus, have increased dramatically. However, little is known about the pathogenicity or the mechanisms of brucella and influenza coinfections. (2) Methods: To clarify the interventions in the early stages of lung damage due to brucella and influenza coinfections, we evaluated the effect of the coinfection on disease progression and mortality using a coinfection model in WT mice and NLRP6^−/−^ mice, and we verified the function of NLRP6 in infection and proinflammation. (3) Results: The coinfection induced significant respiratory symptoms, weight loss, and a high mortality rate in WT mice. Influenza in the coinfection group significantly increased brucella proliferation in a synergistic manner. Meanwhile, a histological examination showed severe lung tissue destruction and excessive inflammatory responses in coinfected WT animals, and the expression of NLRP6 and IL-18 was dramatically increased in the lung tissues. Furthermore, NLRP6 deletion attenuated lung injuries and inflammation, a reduced bacterial load, and decreased IL-18 protein expression. (4) Conclusions: Our findings indicated that NLRP6 plays a critical role and might be a promising potential therapeutic target for brucella–influenza coinfections.

## 1. Introduction

Brucellosis is a zoonotic disease caused by the Brucella spp., a Gram-negative facultative intracellular bacterium which is transmissible via multiple routes [1], and the inhalation of contaminated aerosols is a common route of transmission [2]. With more than 500,000 new cases each year, human brucellosis is still the most widespread zoonotic disease in the world [3,4]. It is also a major contributor to travel-related morbidity and is linked with significant residual impairments. Due to its heterogeneous and poorly specific symptoms [5], brucellosis is often under-diagnosed or misdiagnosed [6,7]. Brucellosis without treatment progresses to a disabling chronic disease with severe complications, such as central nervous system (CNS) affectations, osteomyelitis, keratitis, and endocarditis, and once the brucellosis progresses to a chronic phase, it is difficult to cure [8,9]. There is still no safe and effective vaccination or specific medicine available for humans [10]. Brucellosis is the most prevalent laboratory-acquired infection in the world, and its airborne transmission has been linked to the majority of cases [11,12]. Although brucella frequently infects the host through the respiratory, the majority of the investigations on brucella’s pathogenesis and immunity have been performed using intraperitoneal infection models in animals and, to a lesser extent, oral infection models.

Influenza creates annual epidemics that infect up to twenty percent of the population and that result in substantial morbidity and mortality [13]. Influenza–bacterial coinfections, which are associated with both pandemic and seasonal influenza virus illnesses, are a major cause of morbidity and mortality [14,15,16,17]. According to lung tissue samples from the 1918 influenza pandemic, the bulk of the estimated 20–60 million fatalities were from bacterial infections rather than the virus itself [18]. During seasonal epidemics, Influenza–bacterial coinfections are connected with increasing hospital admissions, symptom severity, and mortality [19,20,21,22]. Clinically, the identification of coinfected pathogens enables clinicians to commence an appropriate antibiotic therapy and to enhance patient outcomes [23]. However, a previous study has cast doubt on the efficacy of antibiotics in treating coinfections [24,25]. Meanwhile, there is still no effective target for Influenza–bacterial coinfections, and the potential *molecular* mechanism underpinning the synergistic pathogenic operations of the two pathogens remains unclear.

NLRP6, a member of the NLR family, regulates the host’s defense against pathogens, such as bacteria, viruses, and parasites [26,27,28,29,30]. During the infection of the lungs and intestines with different microbes, NLRP6 exhibits a dissimilar effect. In the defense against Streptococcus pneumoniae and Staphylococcus aureus infections, NLRP6 has a negative effect on the resistance to the infection [27,28] in which the expression of NLRP6 can cause inflammation and tissue damage. However, NLRP6 plays a protective role during Klebsiella pneumoniae lung infections [26]. As previously reported by Rungue et al., NLRP6 plays a damaging role in the intestinal phase of brucella infections [29]. However, the role of NLRP6 in brucella abortus lung infections has not been reported. In addition, given that NLRP6 has an antiviral role in the intestines, it would be interesting to see if NLRP6 plays a protective or negative role in pulmonary host defense during Influenza infections [30]. In addition, no experimental evidence is available concerning Influenza A virus and brucella coinfections.

In this study, we determined the effect of NLRP6 and its downstream molecules using an aerosol coinfection model with two mouse strains derived from WT (C57BL/6N) and NLRP6^−/−^ mice. Our findings indicated that NLRP6^−/−^ mice were more resistant to Brucella *Suis* and Influenza coinfections since there was a lower systemic inflammation and bacterial load in the lung tissue after coinfection. Moreover, the decreased GSDMD and IL-18 expression in NLRP6-deficient mice may partly explain their reduced inflammatory response. We hypothesized that NLRP6 participates in the pathogenesis of the Influenza–bacterial coinfection and may be a useful therapeutic target.

## 2. Materials and Methods

### 2.1. Mice

The wild-type mice were purchased from vital river, Beijing. Four-week-old female C57BL/6N mice were used. NLRP6-deficient (NLRP6^−/−^) mice were purchased from Cyagen Biosciences, Suzhou, and were backcrossed for at least six generations of C57BL/6 mice in the animal facility of the Institute of Microbiology. Mice were kept in special pathogen-free (SPF) environments. The Beijing Laboratory Animal Welfare and Ethical Guidelines of the Beijing Administration Committee of Laboratory Animals were followed in all animal experiments, which were authorized by the Research Ethics Committee of the Chinese Academy of Sciences. All mice were kept in a barrier-free environment with unrestricted access to food and water.

### 2.2. Bacterial and Virus Strains

Brucella *Suis* strain S2 (CVCC70502) was purchased from Chongqing AoLong. S2 freeze-dried powder was maintained at −20 °C. The bacterial counts were enumerated on TSA media (5% CO_2_ at 37 °C for 72–96 h). The number of bacteria was counted with standard bacterial colony count protocol.

At the Beijing Institute of Microbiology and Epidemiology in Beijing, influenza A/PR/8/1934 (H1N1) was rescued from the State Key Laboratory of Pathogen and Biosecurity. It was then cultured in the allantoic cavities of 9-day-old specific-pathogen-free embryonic eggs and incubated for 48–72 h at 37 °C. After being collected, processed, and kept at −80 °C, the allantoic fluid.

It was recommended that work with S2 and H1N1 should be conducted in a *biosafety level 2* plus facility following *level 2 plus* safety measures.

### 2.3. Virus Titration

Utilizing the median tissue culture infective dosage, IAV was titrated in MDCK cells. After being planted in 96-well plates for 18 h, MDCK cells were removed, and the monolayers of cells were then washed with PBS and injected with five replicates of each 10-fold serially diluted virus for two hours at 37 °C with 5% CO_2_. The cells were then washed three times with PBS after the supernatant was removed, and DMEM containing 0.6% low-melting-point agarose and 1 mg/mL of L-(tosylamide-2-phenyl) ethyl chloromethyl ketone (TPCK)-treated trypsin was then added. IAV titers (108.4 TCID50/mL) were measured after incubation for 72 h at 37 °C with 5% CO_2_.

### 2.4. IAV and Brucella Coinfection In Vivo

Female, 4-week-old C57BL/6 mice of both the wild-type and NLRP6^−/−^ genotypes were anesthetized before receiving an intranasal injection of 10^7^ Brucella *Suis* strain S2 in 50 mL of PBS. Mice in the coinfection group were sedated and given an intranasal IAV injection 12 h later (10^3^ TCID_50_). Body weights and any deaths in mice were determined daily for 14 consecutive days after inoculation. In the case of the number of surviving animals being less than 3 in any group (statistical analysis could not be performed), the weight measurement was stopped.

### 2.5. Plate Colony Counting

The mice were sacrificed 72 h post coinfection, and the lungs were homogenized in 1 mL of PBS. Lung tissues were ground by a homogenizer at 120 Hz for 4 min. Brucella loads were determined with serial dilutions of the samples on TSA plates.

### 2.6. RNA Extraction, cDNA Synthesis, and qPCR Analysis

Total RNA was extracted from lung tissue homogenate with the RNAprep Pure Tissue Kit (Tiangen) according to the manufacturer’s instructions. cDNA was synthesized from 10 mg of total RNA using First-strand cDNA Synthesis Supermix (Transcript) according to the manufacturer’s instructions. Relative gene expression was analyzed with qPCR using PowerUp™ SYBR™ Green Master Mix (Thermo Fisher). The primers are listed in Appendix A. The Ct values were generated from an ABI 7500. The expression of target genes was normalized to that of GAPDH to calculate ΔCT. The ΔCT was used to find the relative expression of target genes according to the following formula:

relative expression = 2−ΔΔCT, where ΔΔCT = ΔCT of target genes in experimental condition − ΔCT of target gene under control condition.

### 2.7. Lung Injury Severity Scoring

The severity of the lung injury was analyzed in a blinded manner with the grader unaware of the study group being reviewed. The lung histopathological changes were assessed by the four identifiable pathologic processes: (1) alveolar wall fracture, (2) alveolar fusion, (3) inflammatory cell infiltration and alveolar hemorrhage, and (4) thickness of the alveolar wall. The scores of 0 to 4 were defined as lower than 25%, 25–50%, 50–75%, and higher than 75% lung involvement, respectively, to represent normal lungs.

### 2.8. Western-Blot

The lung tissue homogenate was collected then freeze-thawed repeatedly 3–5 times using liquid nitrogen. After being centrifuged at 5000× *g* for 7 min, the supernatant was protein extracts. Sample protein concentration was determined with the BCA Protein Assay Kit (Genstar). The supernatants were then divided according to the sample protein concentration. Sample loading buffer was added to the supernatants and boiled for 5 min. Samples were then centrifuged, and the supernatant was removed. Approximately 20 μg of sample supernatant was loaded per lane, separated on a precast polyacrylamide Bis-Tris gel with a 4–12% gradient (Solarbio), and transferred onto a nitrocellulose membrane. The membranes were blocked in 10% milk/TBS-T buffer for 1 h at RT and incubated overnight with the following antibodies: rabbit anti-NLRP6 polyclonal antibody (1:2000, Immunoway), rabbit anti-GSDMD polyclonal antibody (1:5000, Proteintech), and mouse anti-GAPDH monoclonal antibody (1:10,000, Proteintech). Membranes were incubated with fluorescently labeled secondary antibodies (1:10,000, Proteintech) at RT for 1 h. The protein bands were detected with the Odyssey^®^ infrared imaging system (LI-COR Biosciences). The relative expression of each target gene was obtained using the following formula: 

relative expression of gene = (density of the gene band)/(density of β-actin band).

### 2.9. ELISA

Sample supernatant was extracted and determined from lung tissue homogenate with the Mouse TNF-α, IL-1β, and IL-18 Cytokine ELISA Kit (elabscience) according to the manufacturer’s instructions. Details are described in the Appendix A.

### 2.10. Quantification and Statistical Analysis

Statistical analyses were performed using Prism 9.0. Data were presented as the mean values ± SD or SEM. Comparisons between two groups were performed using the two-tailed Student’s *t*-test. One-way ANOVA (analysis of variance) was used to compare three or more groups. *p* < 0.05 was considered significant, and Kaplan–Meier survival curves were analyzed to determine statistical significance with the log-rank test. All scripts of statistical analyses were uploaded and included in the Appendix A.

## 3. Results

### 3.1. Coinfection Caused Acute Lung Injury, which Can Lead to Fatal Conditions

To construct the animal model of coinfections with brucella and influenza, mice were infected with the sublethal doses of the brucella S2 strain and the influenza H1N1 PR8 strain through the respiratory tract. As showcased in Figure 1b, there was a progressive decrease in the body weight of the coinfection groups after the challenge. The single infection mice showed obvious body weight loss and then gradually recovered over 1 week. The percentage of the weight changes was significantly different between the groups (*p* < 0.0001). As showcased in Figure 1c, the mortality rate of the coinfection group was found to be the highest on day 10 post challenge. The difference between the groups of infected mice’s survival curves was statistically significant. The brucella count in the S2 + PR8 group was considerably greater than that in the S2 group, demonstrating that Influenza virus A can promote brucella proliferation in the lungs (Figure 1d). In contrast, the lung viral burden of the S2 + PR8 group was considerably lower than that of the PR8 group, suggesting that Brucella can impede Influenza virus replication (Figure 1e). There were extensive alveolar wall fractures, alveolar fusion, and thickened alveolar walls as well as a large amount of inflammatory cell infiltration (Figure 1f). Corresponding to this, the S2 and PR8 group’s mice tended to have milder symptoms. The results of the pathological scoring could be corroborated with the above results (Figure 1g). In addition, as shown in Figure 1H, the coinfection led to a significant increase in TNF in the lung tissue.

### 3.2. Coinfection Caused an Increase in NLRP6 Expression

The development of ALI may be aided by excessive S2 growth as a result of the coinfection. Thus, our following experiments focused on the mechanisms of bacterial growth and proliferation. A previous study showed that NLRP6 is associated with some common respiratory pathogenic bacteria [26,27,28]. In our study, the relative mRNA expression of NLRP6 was found to be significantly upregulated in the S2 group (Figure 2a). In the PR8 group, there was a small but not statistically significant upregulation in NLRP6 (Figure 2a). However, NLRP6 was dramatically upregulated in the S2 + PR8 group, and the upregulation was greater than the sum of the other two groups (Figure 2a). Moreover, NLRP6 protein expression showed a similar trend (Figure 2c). Therefore, the combination deployment was thought to have a synergistic impact, generating a significantly greater NLRP6 expression than either pathogen alone.

### 3.3. Coinfection Caused an Increase in IL-18 Expression

We further detected the GSDMD mRNA and protein expression. In the S2 group, there was a small but not statistically significant increase in the GSDMD mRNA (Figure 2b). In the PR8 group, its expression was not significantly altered (Figure 2b). However, GSDMD was dramatically upregulated in the S2 + PR8 group, and the upregulation was greater than the sum of the other two groups (Figure 2b). Moreover, GSDMD protein expression showed a similar trend (Figure 2d). This indicated that the coinfection with S2 and PR8 could significantly boost GSDMD expression, which consisted in robust inflammation in the lungs of the coinfection mice. Furthermore, GSDMD’s overall trend was equivalent to that of NLRP6. As a result, we hypothesized that NLRP6 might be involved in regulating GSDMD expression during coinfection, and, through augmenting GSDMD, NLRP6 induced strong inflammation and lung damage.

Compared with the control group, the relative mRNA expression of IL-1β was found to have a slight but not statistically significant upregulation in the three infection groups, and no differences were observed between these groups (Figure 2e). Similarly, although there was a considerable increase in these groups compared to the control group, the cytokine content of IL-1β did not change between the infection and coinfection groups (Figure 2f). On the contrary, IL-18 mRNA expression was shown to be considerably increased in the three infection groups (Figure 2g), with statistically significant differences between the groups. In addition, the IL-18 cytokine content followed a similar pattern (Figure 2h). It was suggested that IL-18 played an essential role in B. *Suis* and IAV coinfections. Furthermore, the overall trend of IL-18 matched that of NLRP6 and GSDMD. Therefore, we speculated that NLRP6 may potentially be involved in regulating GSDMD and IL-18 expression during B. *Suis* and IAV coinfections.

### 3.4. The Involvement of NLRP6 in Brucella and/or Influenza Virus Infections

To confirm the function of NLRP6 in B. *Suis* and IAV coinfections in the lung, we established a NLRP6^−/−^ coinfection mouse model with the same set-ups and acquisition conditions as the previous study. Compared with the WT groups, the bacterial load of the NLRP6^−/−^ mice decreased significantly in the S2 + PR8 group and the S2 group, respectively (Figure 3a). Thus, it was believed that NLRP6 promoted B. *Suis* proliferation. Moreover, in NLRP6^−/−^ mice, the bacterial count of the S2 + PR8 group was still higher than that of the S2 group. Even in the absence of NLRP6, PR8 was thought to enhance S2 proliferation. Therefore, there must be other causes for IAV-induced S2 proliferation. The NLRP6^−/−^ mice showed a less severe inflammatory reaction and tissue injury (Figure 3c,d) when compared to the WT mice. This indicated that an infection with B. *Suis* and/or IAV could cause NLRP6 to be activated in the lungs of mice, which could exert a proinflammatory effect.

### 3.5. NLRP6 Mediated the Upregulation of IL-18 and IL-1β

Compared with the WT groups, the RT-qPCR and ELISA showed a significantly reduction in IL-18 expression in the NLRP6^−/−^ groups (Figure 4c,d). It was suggested that NLRP6 mediated the elevation of IL-18 in B. *Suis* and/or IAV-infected lung tissues. In addition, the differences between the WT groups were highly significant (Figure 2f,g); however, in the NLRP6^−/−^ mice, these differences were absent. In conclusion, with the coinfection with B. *Suis* and IAV, NLRP6 can increase IL-18, resulting in potent proinflammatory effects that can harm the mouse lung tissue. Moreover, there was no difference in IL-1β mRNA expression in the WT and NLRP6^−/−^ groups (Figure 4a). But an ELISA showed a significantly reduction in the expression of IL-1β cytokines in the NLRP6^−/−^ groups (Figure 4b). This was concordant with prior studies [28]. NLRP6 was involved in IL-1β maturation and secretion but not in IL-1β transcription induction.

## 4. Discussion

Brucellosis is crippling but is rarely deadly. Acute sickness in people is characterized by nonpathognomonic clinical symptoms, such as sadness, myalgia, arthralgia, splenomegaly, and undulant fever [31]. Most patients are easily misdiagnosed as having a cold during their first visit in the Department of Respiration [32]. In China, brucellosis has been reported in all of its provinces and is endemic in 25 provinces (or autonomous regions) [33]. Although the risk posed by bacterial coinfections in influenza patients is well understood [14], there are no effective strategies to reduce the severity and mortality of these coinfections. Recent observations in China indicate a recurrence of brucellosis. Although comparatively fewer reports have been published on brucella and Influenza coinfections, our data exists to support their damage role in accelerating lung injuries. Therefore, it is essential to clarify the characteristics of these coinfections and to explore coping mechanisms.

Firstly, taking into consideration that severe pneumonia is a common respiratory disease in children and that it is also one of the main causes of death in hospitalized children under 5 years of age, 4-week-old C57BL/6 mice were used as animal models in our study. Previous studies showed that Influenza-infected mice exhibit peak susceptibility for bacterial coinfection at the peak of lung injury, roughly 7 days post Influenza [34,35]. However, the direct synergistic pathogenic effects of the bacteria and virus were a major concern for our study. Thus, we infected the mice twice in a short interval to reduce the disturbances of lung injuries, which can result in heightened susceptibility.

In our study, we confirmed that respiratory and systemic symptoms, nonprotective inflammation, lung tissue damage, and a high mortality rate were observed in the WT mice during the early stages of the brucella and Influenza coinfection. This synergistic effect was consistent with what happens when Influenza and S. pneumoniae or S. aureus work together [36,37]. In the coinfection group, the increase in the brucella count was 30 times greater than that in the solo S2 group. This might be one of the causes of the higher mortality rate and severity with the coinfection challenge. According to Bai’s research, an H1N1 infection enhanced the proliferation and adherence of Gram-positive Streptococcus pneumoniae, which is in line with our findings [38]. Therefore, coinfections with Brucella and Influenza could result in significant adverse effects in humans. If a host is already infected with brucella or IAV, it is necessary to prevent them from being exposed to the other pathogen. Those working in animal husbandry in particular should be aware of the importance of occupational exposure protection.

NLRP6 participates in the host defense against pathogens, such as bacteria, viruses, and parasites. During the infection of the lungs and intestines with different microbes, NLRP6 exhibited a dissimilar effect. Rungue et al. reported that NLRP6 plays a damaging role in the intestinal phase of Brucella infections [29]. In comparison to the WT mice, the NLRP6^−/−^ mice were more resistant to the Brucella infection, with a lower CFU in the liver and lower intestinal permeability [29]. However, the function of NLRP6 in Brucella lung infections is unknown. Thus, we thought that too much S2 growth caused by IAV during the coinfection would be linked to too much NLRP6 expression. Our data indicated that the B. *Suis* and IAV solo infections generated a slight elevation in NLRP6 expression, but the B. *Suis* and IAV coinfection generated the synergistic effect of NLRP6 overexpression. In addition, the NLRP6^−/−^ mice had a lower bacterial load in the lung tissue after the coinfection. It was suggested that NLRP6 hurt the pulmonary host anti-Brucella defense when the two infections occurred at the same time.

Moreover, the coinfection also caused nonprotective inflammation through NLRP6 overexpression. To explore the mechanism of NLRP6, we detected several proinflammatory molecules induced by NLRP6. Previous studies showed that NLRP6 can form a complete NLRP6 inflammasome when it recruits and binds with caspase-1 and/or caspase-11 through the classical or nonclassical pathway [39,40,41]. This inflammasome can cleave GSDMD and can induce the maturation and secretion of IL-1 and IL-18 [42]. A large number of GSDMD pores in the plasma membrane can result in an enormous leakage of the cytosolic contents and inflammatory reactions [43,44]. IL-1β (4.5 nm) and IL-18 (5.0 nm) can be released into the extracellular space through the GSDMD pores (10–15 nm) [40]. This also explains why, just prior to cell cleavage, IL-1 and IL-18 can be observed extracellularly [45]. Some studies have demonstrated that IL-1β and IL-18 can exert proinflammatory effects [46,47]. There is additional evidence that NLRP6 played a role in exacerbating the symptoms of systemic listeriosis by boosting the expression of IL-18. IL-18 treatment in the NLRP6^−/−^ mice restored the mutant mice’s sensitivity to L. monocytogenes infections [48]. Sandip et al. showed that mice with Citrobacter rodentium infections exhibited a high amount of activated IL-18, which was triggered by NLRP6 and caused significant colonic inflammation [49]. Therefore, it is possible that IL-1β and IL-18 production induced by NLRP6 plays another key role in the pathogenic mechanisms of B. *Suis* and IAV coinfections. However, whether NLRP6 signaling is involved in the transcription mechanism of IL-18 and IL-1β remains unclear. In our study, an increase in active IL-18 and IL-1β cytokines was observed in the WT to NLRP6^−/−^ groups (Figure 4b,d). It was suggested that NLRP6 could promote pro–IL-18 and pro-IL-1β into active IL-18 and IL-1β. Furthermore, we found that NLRP6 was involved in the transcription of IL-18 but not in the induction of IL-1β transcription during the coinfection (Figure 4a,c). The coinfection induced the potent expression of IL-18, and the overall trend of IL-18 was equivalent to that of NLRP6. Therefore, we speculated that, with the coinfection with B. *Suis* and IAV, NLRP6 could increase IL-18, resulting in massive proinflammatory effects that could harm the mouse lung tissue. Further validation experiments will be performed in anti-IL-18 mAb and IL-18^−/−^ models.

Interestingly, the RNA expression of PR8 showed a statistically significant upregulation in the NLRP6^−/−^ coinfection group, suggesting that IAV increased due to S2 synergy (data not shown). However, our previous data demonstrated that Brucella can impede IAV replication in WT mice. Previously, Wang et al. reported that NLRP6 works with Dhx15 as a viral RNA sensor to trigger ISGs and that this effect is particularly essential to antiviral signaling in the intestine [50]. Therefore, we speculated that Brucella lung infections could increase the Influenza load in the coinfected group. However, in vivo, B. *Suis* could indirectly impede IAV replication through NLRP6 upregulation. This effect has not been recorded, and further experimental verification is required.

This work expanded on the clinical spectrum of infections by examining coinfections with the Influenza virus and Brucella in the lung tissue. Further research was done on NLRP6 and IL-18, which are possible targets for intervention since they can negatively impact ALI. There were also some limitations in this study. First, since this was a cross-sectional study, it had the usual limitations of a cross-sectional analysis. Second, even in the NLRP6 knockout mice, the bacteria count of the S2 + PR8 group was still greater than that of the solo S2 group. It was suggested that there were definitely other possible mechanisms for the conferment of B. *Suis* proliferation promoted by IAV. Further studies are warranted to comprehensively understand the underlying mechanisms.

In conclusion, compared with the sole infections, a greater bacteria count, inflammatory reaction, and high mortality were exhibited in the early stages of the coinfection with brucella and influenza. Additionally, the coinfection caused NLRP6 overexpression, which plays a role in proinflammation in the lung tissue of mice. IL-18, which is a powerful inflammatory molecule, could be increased by NLRP6 in the lungs because of the coinfection. Thus, it looked like NLRP6 and IL-18 could be promising molecules to target early on in this coinfection to stop an overactive inflammatory response and to stop damage to the tissues.

## Figures and Tables

**Figure 1 jpm-12-02063-f001:**
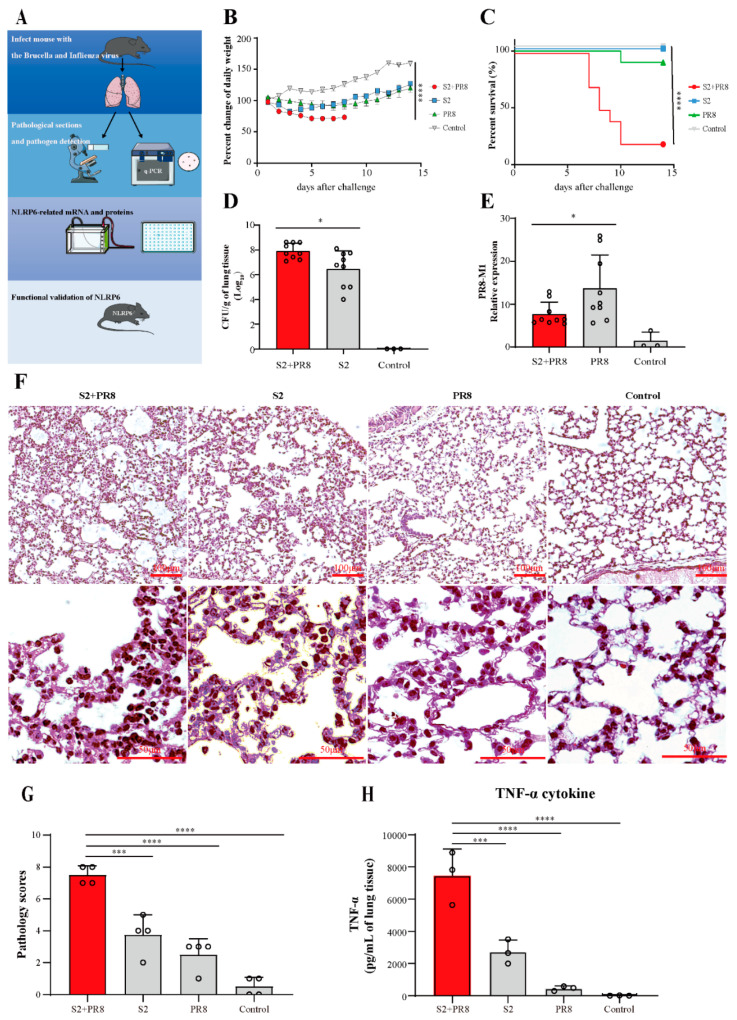
Coinfection caused massive Brucella proliferation and lethal lung tissue damage. (**A**) Experimental flow chart. Mice were infected with Brucella and Influenza through the respiratory tract. Mice were sacrificed, and lungs were collected via sterile dissection 3 days after infection. Lung pathology, viral/bacterial load, and inflammation-associated molecules and cytokines were detected for analysis of the proinflammatory mechanism of coinfection. Finally, the function of NLRP6 using NLRP6^−/−^ mice was validated. (**B**) Weight changes in mice. Results expressed as mean ± standard deviation. The data were analyzed using repeated measures ANOVA. (**C**) Coinfection mice mortality rate increased: comparison of survival curve analysis. (**D**,**E**) The Brucella load (log_10_) increased in the coinfected group, while the viral load decreased when compared to single infection. Two-tailed unpaired Student’s *t*-test for two groups was performed. (**F**) Coinfection caused massive inflammation and severe tissue damage. Pathological sections of lung tissue were observed under a light microscope at 3 days post infection. (With magnification of 10× and 40×). (**G**) Graph showing pathology score of the lung tissues after the infection. Two-tailed unpaired Student’s *t*-test for two groups. (**H**) The TNF-α cytokine was measured with ELISA. Results expressed as mean ± standard; * *p* < 0.05; *** *p* < 0.001; and **** *p* < 0.0001.

**Figure 2 jpm-12-02063-f002:**
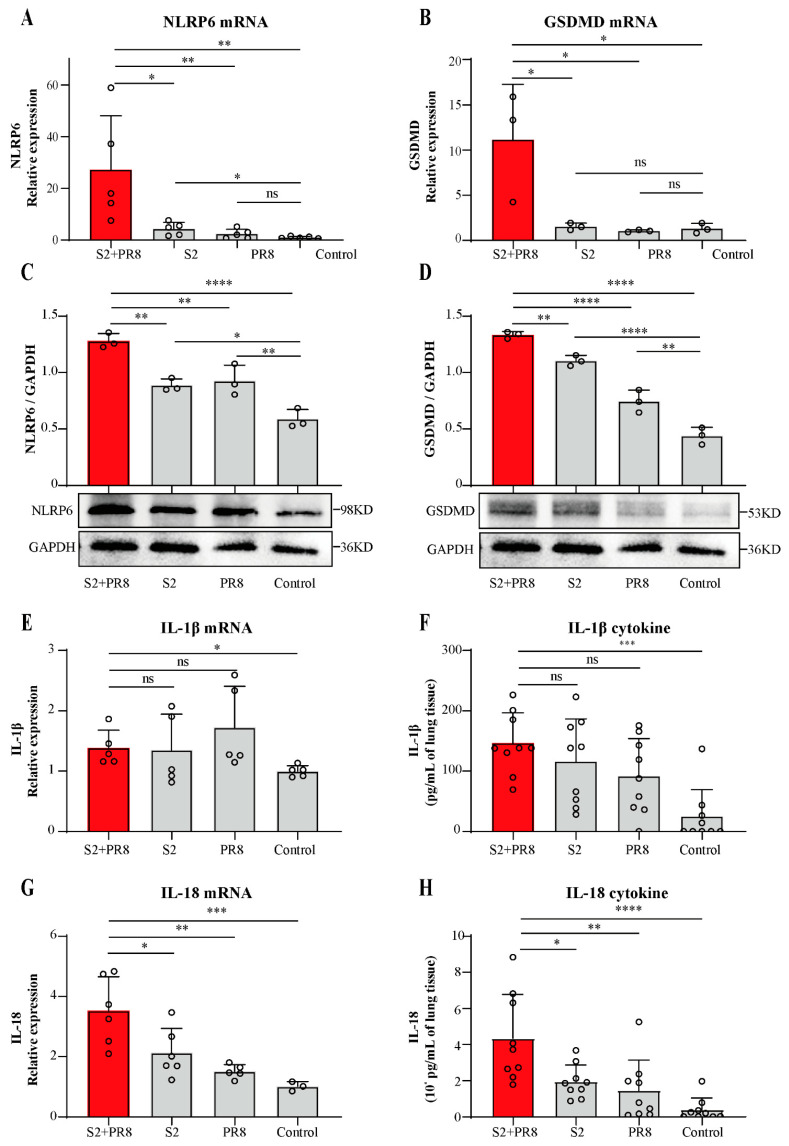
The expression levels of NLRP6, GSDMD, IL-1β, and IL-18 in WT mice. (**A**–**D**) The overexpression of NLRP6 and GSDMD was caused by coinfection in the lungs. The mRNA expression was measured with qPCR, and the protein expression was detected with Western blot. Two-tailed unpaired Student’s *t*-test was used to compare two groups, and one-way ANOVA was used to compare three groups or more. (**E**–**H**) The expression of IL-1β and the overexpression of IL-18 were caused by coinfection in the lungs. The mRNA expression was measured with qPCR, and the cytokines were measured with ELISA. Two-tailed unpaired Student’s *t*-test were used to compare two groups, and one-way ANOVA was used to compare three groups or more. Results expressed as mean ± standard; * *p* < 0.05; ** *p* < 0.01; *** *p* < 0.001; and **** *p* < 0.0001.

**Figure 3 jpm-12-02063-f003:**
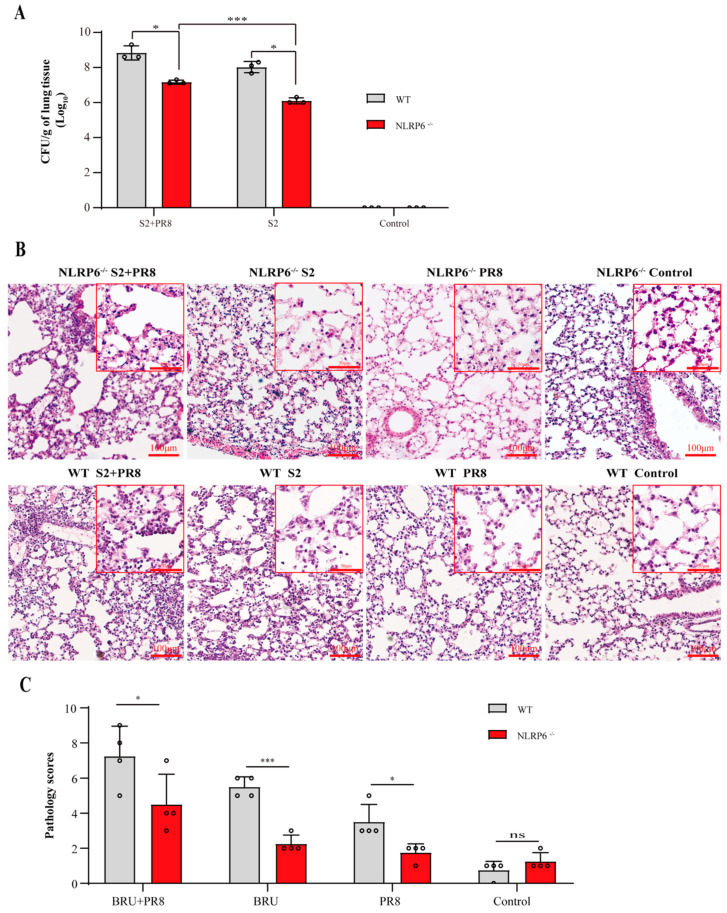
NLRP6^−/−^ mice were more resistant to coinfection than WT mice. (**A**) The Brucella count (log_10_) of NLRP6^−/−^ was more significantly decreased than that of WT mice. Two-tailed unpaired Student’s *t*-test was used to compare two groups. (**B**) The levels of inflammation and tissue damage in NLRP6^−/−^ mice were milder than those in WT mice. Pathological sections of lung tissue were observed under a light microscope 3 days post infection. (With magnification of 10× and 40×). (**C**) Graph showing pathology score of the lung tissues after the infection. Two-tailed unpaired Student’s *t*-test was used to compare two groups. Results expressed as mean ± standard; * *p* < 0.05; *** *p* < 0.001.

**Figure 4 jpm-12-02063-f004:**
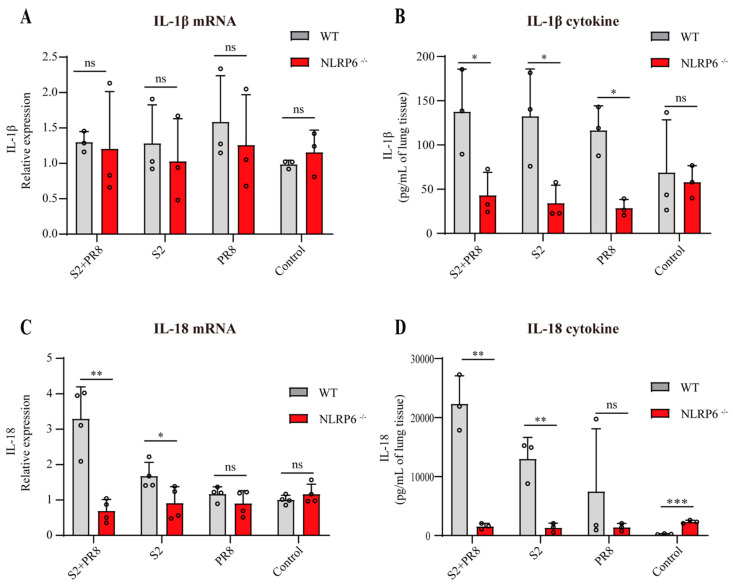
The expression levels of IL-1β and IL-18 in NLRP6 deleted mice. (**A**,**B**) The mRNA of IL-1β was not differentially expressed. The cytokine IL-1β was downregulated in NLRP6^−/−^ mice. Two-tailed unpaired Student’s *t*-test was used to compare two groups. (**C**,**D**) The mRNA and protein of IL-18 were differentially expressed. Two-tailed unpaired Student’s *t*-test was used to compare two groups. Results expressed as mean ± standard; * *p* < 0.05; ** *p* < 0.01; *** *p* < 0.001.

## Data Availability

The data presented in this study are available in the Appendix A.

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
