# Peer review of "NLRP6 Induces Lung Injury and Inflammation Early in Brucella and Influenza Coinfection"

_jpm, 2022, doi:10.3390/jpm12122063_

Round 1

Reviewer 1 Report

Studying coinfection is an important topic that helps to devise efficient treatment protocols for rare clinical cases. The authors appropriately apply statistical tests to support their findings and include raw data necessary for reproducing the results. Overall, the work is of good quality. My only suggestions would be to describe in more detail the way relative gene expression was calculated and also to include scripts for reproducing t-test and ANOVA results (any platform is fine).

Author Response

Response to Reviewer 1 Comments

Point 1: Studying coinfection is an important topic that helps to devise efficient treatment protocols for rare clinical cases. The authors appropriately apply statistical tests to support their findings and include raw data necessary for reproducing the results. Overall, the work is of good quality. My only suggestions would be to describe in more detail the way relative gene expression was calculated and also to include scripts for reproducing t-test and ANOVA results (any platform is fine).

Response 1: Thanks to your professional advice, we have added detailed calculations in the methods section of the article. In addition, the script file for the statistical analysis (Original data.pzfx) was uploaded via e-mail.

Reviewer 2 Report

The authors have conducted an animal study looking into NLRP6 associated lung injury and inflammation in mice coinfected with strains of influenza and brucellosis. The conclusions are reflective of the presented evidence. No obvious flaws or weakness in the manuscript.

Author Response

Response to Reviewer 2 Comments

Point 1: The authors have conducted an animal study looking into NLRP6 associated lung injury and inflammation in mice coinfected with strains of influenza and brucellosis. The conclusions are reflective of the presented evidence. No obvious flaws or weakness in the manuscript.

Response 1: Thank you for your recognition and thank you from the bottom of my heart for all your hard work!

Reviewer 3 Report

I congratulate with the authors, the work is a fine example of a good scientific work. Elegant, well structured and scientifically valid and very interesting. I find it very interesting since the co-infections between influenza viruses and bacteria are little investigated. I have only a few minor considerations.

Minor questions:

• In materials and methods for gene expression experiments (page 3, lines 132-134) G6PDH was used for the reference gene, if another houskeeping was used (eg beta-globin) this would have strengthened the data. Since the ct of the investigated genes are not always similar to those of the reference genes, therefore having at least two different houskeeping would be better. Was another reference gene included in the experiments? if it should be specified.

• In the section Materials and methods for ELISA experiments (page 4, lines 159-161) I would suggest the authors add the list of cytokines assayed.

• In the data on the WB analysis on page 7 (figure 2C) the housekeeping picture is not perfect. The first band on the left seems a little more intense than the last band on the right (in the G6PDH line). However I am sure that the authors have certainly loaded the same amount of sample in each well, from my personal experience I know well that WB experiments are never perfect.

I would suggest adding a normalization graph of the densitometric data next to the photo of WB's experiment, to graphically show the increased expression of NLRP6.

• Since it is known that cytokines such as TNF-α are produced in inflammatory responses following tissue damage and this is known to induce NLRP6 transcription, have the TNF-α levels been measured in lung tissues of mice? if yes, the data should be added.

• IL-18 activity is regulated by an inhibitor which is IL-18BP, have IL18PB levels been evaluated in the mice studied ? if yes, the data should be added to enrich and give more completeness to the overall data.

• In some tissues NLRP3 Activation has opposite roles to NLRP6. Was the activation/expression of NLPR3 evaluated in mice ? if yes it should be added because it would enrich the study.

Author Response

Response to Reviewer 3 Comments

I congratulate with the authors, the work is a fine example of a good scientific work. Elegant, well structured and scientifically valid and very interesting. I find it very interesting since the co-infections between influenza viruses and bacteria are little investigated. I have only a few minor considerations.

Point 1: In materials and methods for gene expression experiments (page 3, lines 132-134) G6PDH was used for the reference gene, if another houskeeping was used (eg beta-globin) this would have strengthened the data. Since the ct of the investigated genes are not always similar to those of the reference genes, therefore having at least two different houskeeping would be better. Was another reference gene included in the experiments? if it should be specified.

Response 1: Thank you for your suggestion. As stated in your professional review, we conducted replicate experiments using GAPDH and β-actin as reference gene separately to strengthen the credibility of the results. The raw experimental data showed similar results and trends, so we only displayed the results of one of them in the manuscript and figures.

Point 2: In the section Materials and methods for ELISA experiments (page 4, lines 159-161) I would suggest the authors add the list of cytokines assayed.

Response 2: We would like to and have placed abbreviated information on cytokines in the corresponding position and added the list to the supplementary material table.

Point 3: In the data on the WB analysis on page 7 (figure 2C) the housekeeping picture is not perfect. The first band on the left seems a little more intense than the last band on the right (in the G6PDH line). However I am sure that the authors have certainly loaded the same amount of sample in each well, from my personal experience I know well that WB experiments are never perfect. I would suggest adding a normalization graph of the densitometric data next to the photo of WB's experiment, to graphically show the increased expression of NLRP6.

Response 3: Thank you very much for your understanding, the picture has been added in the corresponding position.

Point 4: Since it is known that cytokines such as TNF-α are produced in inflammatory responses following tissue damage and this is known to induce NLRP6 transcription, have the TNF-α levels been measured in lung tissues of mice? if yes, the data should be added.

Response 4: OK, cytokine assay results with TNF-α added.

Point 5: IL-18 activity is regulated by an inhibitor which is IL-18BP, have IL18PB levels been evaluated in the mice studied ? if yes, the data should be added to enrich and give more completeness to the overall data.

Response 5: Thank you for your enthusiastic reminder! Unfortunately, we overlooked this point in the initial experimental design, and we will focus on its changes and effects in the subsequent study.

Point 6: In some tissues NLRP3 Activation has opposite roles to NLRP6. Was the activation/expression of NLPR3 evaluated in mice ? if yes it should be added because it would enrich the study.

Response 6: Your comments are very professional and we considered similar issues at first, but in the end we thought that as a preliminary exploration, the experimental design and expected results should be more concise to avoid ambiguous doubts and they may not be conducive for us to explain the function of NLRP6. Therefore, we only conducted functional studies of a single molecule. But as you said, this interaction will be the next step of our study if we want to fully resolve the function of a molecule.